# The Impact of COVID-19 on Canada’s Food Literacy: Results of a Cross-National Survey

**DOI:** 10.3390/ijerph18105485

**Published:** 2021-05-20

**Authors:** Sylvain Charlebois, Janet Music, Shannon Faires

**Affiliations:** Agri-Food Analytics Lab, Dalhouise University, Halifax, NS B3H 4R2, Canada; janet.music@dal.ca (J.M.); shannon.faires@dal.ca (S.F.)

**Keywords:** food literacy, crisis, food security, cooking, well-being

## Abstract

Several studies of food literacy emphasise the acquisition of critical knowledge over context. This evaluation looks at how COVID-19 impacted food literacy in a country affected by the global pandemic. To our knowledge, there has been no systematic research that would allow a better understanding of the impact of uncertainty or enhanced perceived risks generated by a global crisis on the prevalence of household food literacy. This study looks at food literacy from a perceptive of how an event that has domesticated many of them can alter knowledge and the relationship people have with food. A cross-national survey including 10,004 Canadians was conducted ten months after the start of the pandemic. Results show that Canadians have learned new recipes. Canadians have also taken up gardening and have relied on several sources to gather information. This study provides some evidence that Canadians have become more food literate because of the COVID-19 pandemic, but less significantly than anticipated. Practical and policy implications are presented as well as some future research directions.

## 1. Introduction

Food is a central part of our everyday lives, and how we connect with food will vary between nations, cultures, households, and individuals. Food literacy is seen as a vital element in improving the increasing levels of food insecurity or perceived food insufficiency [1]. Food knowledge and competencies aiming at healthier behaviours can also be impacted by food literacy [2,3].

Food literacy is related to an individual’s capacity to make feasible food decisions that balance food needs using available resources [4,5]. A person should be able to assess the quality of food options, including how impactful food choices are to the welfare and well-being of communities. Having food literacy means that a person can make practical food decisions that balance food needs using accessible personal resources [6].

When examining the discourse surrounding food literacy, cooking often materialises as a dominant theme [7]. Embedded in the concept of food literacy is health and nutrition literacy, which are topics that have been well researched over the years. In other words, food literacy is one of the aims of food education. Others include kitchen-cooking literacy, nutrition literacy or health literacy [8,9].

To our knowledge, there has been no systematic research that would allow a better understanding of the impact of a global crisis on the prevalence of household food literacy. The COVID-19 pandemic has influenced most consumers to make different choices. These choices have been largely impacted by how they perceived risks. This is probably because measures of food literacy used in a crisis context are seldom found together in datasets. The global coronavirus offers a unique food literacy context. To address this gap in the literature, we collected data on a nationally representative sample of Canadians about both food literacy and how it affects financial literacy [10,11].

Few studies currently document the heightened and uneven vulnerabilities to food insecurity during the COVID-19 pandemic [12]. This study looks at how a global pandemic, the COVID-19 crisis, may have impacted food literacy in Canada. The country has been impacted as much as any other western world countries. The first COVID-19 case in Canada was discovered on 20 January 2020 [13,14]. The virus spread through most provinces by the end of March as the first wave of lockdowns were enforced. Canada experienced a second wave of the virus in the Fall, and throughout Winter 2021, as the number of daily cases reached record levels [15], the second wave of lockdowns was enforced. Measures varied greatly between provinces, but many schools were closed, and people were asked to work from home. By January 2021, it was estimated that about 40% of the Canadian workforce was working from home most of the time [16].

Most consumers want to lead a healthy lifestyle while making proper food decisions but often with limited information or knowledge. The pandemic disrupted the balance between home and professional life for many, and the uncertainty likely became overwhelming [17]. Therefore, to question if such an experience can make a consumer more aware of food systems and/or if they will seek more information is appropriate. We postulate that a change in risk perceptions and lifestyle can trigger more awareness towards food systems. Given this paradox, we aim at understanding how a crisis, which causes enhanced uncertainly, makes one more food literate [18]. The study uses the context of the COVID-19 pandemic in Canada. Previous research has highlighted the need to consider food literacy as a lever towards behavioural change, but very little attention has been given to how context or a sudden environmental certainty generated by a crisis can lead to a change in food literacy.

This paper is organised as follows. First, we present the existing literature on food literacy and information sharing, coupled with a review on risk perceptions in times of uncertainty. This paper explains how a cross-national exploratory survey was designed and executed. Our results are based on a survey module designed by the research team to better understand the relationship between food literacy and socioeconomic groups. After results are highlighted, a discussion on practical and policy implications are presented and suggests how this study can inspire future work in food literacy.

## 2. Food Literacy

In the Western world, consumers have reported that they lack the appropriate information that would help them to be more informed to make adequate decisions regarding different aspects of food-related choices [2]. Food literacy varies between countries and how it is affected by systemic factors such as topography, history, and education. Food literacy holds some significance for a country such as Canada, where food security implies self-reliance on food production at community and national levels. This said, Canada does rely heavily on international trades to remain food secure, which may not create the resiliency against any natural disasters or political problems a country occasionally experiences compared to other, less food-secure countries. Canada is a largely self-sufficient country and one of the richest countries in the world. It has never experienced famine or major food security crises like other countries in the Western World, let alone in the developing world. Since the country’s inception in 1867, Canada has never experienced war in its territory. Canada has experienced a few public health crises over the years, with the Spanish Flu being the most significant one. The shock created by the COVID-19 pandemic, and subsequent public health measures, made the Canadian population more domesticated in a matter of days. As a result, many Canadians experienced confinement for the first time in their lives.

As a result of the pandemic, the importance of food literacy has shifted in Canada [19]. As the COVID-19 pandemic unfolded and continues to, substantial attention has focused on the resiliency of food supply chains in times of uncertainty. Food supply chains have needed to adjust rapidly to demand-side tremors, including unnecessary panic buying and changes in food purchasing patterns by consumers, as well as plan for any supply-side disruptions due to potential labour shortages and disruptions to transportation and supply networks [20]. Restaurants and many foodservice outlets have closed due to public health measures, which triggered massive shocks towards food retailing. What food products have been purchased by consumers have varied greatly throughout the pandemic. At first, non-perishable products were popular. However, as time went on, consumers started to gravitate more towards the periphery of stores where fresh products are located [21]. Food literacy, which involves consumers’ capacity to cook and prepare food at home, may have influenced how and what they purchased throughout the pandemic; however, this has not been measured since the start of the pandemic.

Food literacy is a central component of what consumers can do with food and fundamentally change their relationship with food in general [22]. To assure food literacy, the type of knowledge shared is not only difficult to control, but also difficult to predict and therefore represents a concern to policymakers as it influences public opinion and policy [23].

A growing body of research has documented the significance of food literacy, the characteristics of households that are food illiterate, and the harmful consequences [24]. Food literacy has gained increasing importance in agri-food research throughout the last few decades [25]. Food education is considered as the teaching of knowledge and skills that help people make appropriate decisions for themselves, the environment, and the community [26]. Some of these choices can include cooking, gardening, and how information is sought by individuals [27].

Food literacy is considered a consumer empowering process in which individuals gather inter-related knowledge, skills and behaviours required to prepare, manage, and consume food to meet personal and collective needs and to determine intakes. Enhancing food literacy enables relationships between food, people, and well-being [28].

## 3. Risk Perception and Uncertainty

Environmental uncertainty will impact a population’s capacity to perceive risks [29,30]. Such a powerful sentiment will alter how food secure a population will be or feel. Food insecurity during natural, economic, or public health crises, real or perceived, has a nearly predictable outcome [31,32,33]. While we expect food insecurity to be elevated during these types of events, the magnitude of food insecurity and, in turn, how food insecurity can be perceived by consumers who live in a relatively affluent economy has received relatively little attention [34]

Beyond social vulnerabilities, there are individual-level risks factors related to health that also impact food insecurity [35]. The relationship between food literacy and risk perception has rarely been made. A heightened sense of risk will get someone to limit the risks almost instantaneously. Food literacy may be a method to help one cope and mitigate risks [10,36]. The context will impact how we live and how we make decisions. How people perceive risks will vary based on the information at hand and one’s emotions, and how the information is processed [37].

Some people mentioned that the risks of communicating food-related information in a crisis environment would influence perceptions and behaviour [27]. More specifically, in contrast with normal times when information can be processed more meaningfully, food education and information circulated through different sources cannot entirely be controlled by nutritional specialists and policymakers as information can be distorted [38]. The risks of knowledge distortion are often quite high when consumers operate daily within the constraints of busy lifestyles, but risks are higher in times of uncertainty (Hagedorn et al., 2018). Confinement and lockdowns have made social media more influential in changing perception and how information has been shared broadly [19].

A crisis or a sudden imbalance in people’s daily routines may be a great opportunity for nutritional education to occur, which could, in turn, create a quest to improve healthy eating behaviours and practices [39].

What is also clear in the literature is how the trust of information can impact one’s behaviour. The higher the public trust and evaluation of information, the greater the compliance will be. Differently from other epidemics or crises, the COVID-19 crisis has widespread economic and social consequences; therefore, in most cases, it is impossible to assess risks rationally. The situation with the pandemic is arguably unprecedented from a risk perception point of view [40].

## 4. Methodology

The main data source for this research comes from a survey module we collected in the Caddle panel (CP). CP is a nationally representative Internet panel of respondents 18 years and older who agreed to participate in occasional online surveys with Dalhousie University. Respondents were recruited using a nationally representative sampling frame. Upon joining the panel, respondents complete an initial survey collecting individual socio-demographic information, work history, and household composition information. They are also asked to update their background information each time they log in to respond to a module. Roughly once a week, respondents receive an e-mail with a request to fill out a questionnaire. Response rates average 70–80%. Since 2018, CP has included over 100 surveys on a wide range of topics. For this survey, a structured questionnaire containing questions, mostly close-ended, to ease data processing, minimise variation, and improve the precision of responses was designed. The sample included 54% self-identified females versus 45% self-identified males. A total of 45% of respondents were Millennials (1981–1996), 34% were Gen Xs (1966–1980), and 15% were Boomers (1946–1965). A total of 59% of households had children in them. Information about salaries is shown in Figure 1. No data on race were collected for this survey.

The survey was divided into five sections. We first assessed the pillars of food literacy in terms of importance, health, community, environment, and economy. The second section looked at the number of recipes respondents knew and whether they were self-designed or directed. Recipes and cooking are strong metrics to assess food literacy [41]. For this study, a recipe was defined as a set of at least three steps for preparing a particular dish, including a list of at least three ingredients required. It also looked at the use of new ingredients used since the start of the pandemic. The questionnaire then asked questions about meal and snack management, sources of information trusted and influenced by. The fourth section asked about gardening, another metric to assess food literacy [42]. Finally, the fifth section discusses well-being since the start of the pandemic (Appendix A).

Data were collected from 10,004 respondents in a survey that was fielded in January 2021, about ten months after the start of the pandemic and after lockdowns in Canada. Sample weights were calculated to make the distributions of age, gender, race/ethnicity, education, income, and household size approximate to the distributions in CP, in order to increase the generalisability of the results. The margin of error for the survey, given the sample size, is approximately 1.3%, 19 times out of 20. Any discrepancies in or between totals are due to rounding.

## 5. Results

Given that food literacy can be a coping mechanism to offset these vulnerabilities, this exploratory survey garnered interesting descriptive results. This section presents the results of our analysis of the relationship between food literacy and health perceptions.

When asked about awareness of how food choices can impact aspects of our lives, 70.5% of Canadians believe health is most important, followed by the economy at 52.7%. The environment is third at 28.3%, and community is last at 23.0% (Figure 2).

The survey suggests that 86.7% have heard about the concept of food literacy at some point, but only 39.5% of respondents claimed to know it well enough to explain it. As far as teaching food literacy in schools, 91% of Canadians support it, which is not very surprising.

We also looked at cooking, which is a significant part of how food literate someone can be. A total of 24.3% of Canadians claim that they have prepared all the meals consumed since the beginning of the pandemic. A total of 55.9% of Canadians feel that they have prepared most meals themselves. As we were cooking more, a total of 35.5% of Canadians have learned a new recipe since the start of the pandemic (Figure 3). Given the amount of time many Canadians have spent at home, this figure was lower than expected. The province of Quebec is where most people learned a new recipe since the beginning of the pandemic. In that province, 37.2% learned a new recipe since March 2020. The lowest rates were found in Manitoba and Nova Scotia, where only 30.8% learned a new recipe.

However, when asked about the number of known recipes, the numbers are telling us that Canadians’ knowledge about new recipes has barely changed since the start of the pandemic. Before the pandemic, 8.6% of Canadians did not know one recipe. That percentage dropped to 7.0% in January 2021. Most Canadians now claim to know at least 10 recipes. Before the pandemic, 56.6% of Canadians knew 7 or more recipes. Now, 62.1% of Canadians know 7 recipes or more, which means that the number of Canadians who know 7 recipes or more has increased by 9.7%. Based on these results, the average Canadian knew 6.2 recipes before the pandemic. That ratio slightly increased to 6.7 recipes. Given how more domesticated Canadians have become, we were expecting that ratio to be much higher.

It appears that the Boomers is the generation that knows the most recipes on average. The average Boomer knows 7.6 recipes now versus 7.4 before the pandemic (Table 1). That is the lowest increase of all generations. Canadians who are part of the Generation Z group know the least number of recipes. Before the pandemic, a Gen Z knew on average 4.7 recipes and now would know 5.6 recipes. Generation X followed the Canadian average in terms of the number of known recipes, before the pandemic and now. It is worth noting that Millennials have learned to cook the most during the pandemic. Before the pandemic, the average Millennial knew 4.9 recipes. That number has jumped to 6.0, the highest increase of all generations. Essentially, the pandemic has enticed younger generations to learn more recipes, more so than older generations. The link between age and the number of recipes known remains strong.

Unsurprisingly, salaries are also a determinant when looking at cooking. All income brackets have been impacted by the pandemic, and all groups know more recipes than before the pandemic. Canadians earning more than $75,000 a year tend to know more recipes than people with a lower income. Canadians with an annual income of below $75,000 now know 5.6 recipes on average, versus 7.1 recipes for Canadians earning more.

People tend to teach themselves new recipes before and during the pandemic. Since the start of the pandemic, 38% of Canadians have taught a new recipe to someone they live with, and 37% of Canadians have designed a new recipe for themselves.

A total of 48% of Canadians have used a new ingredient they never used before the start of the pandemic. Spices were the most popular choice for new ingredients by Canadians. A total of 67.5% of Canadians have tried new spices, followed by vegetables at 36.9%, and oils at 27.9% (Figure 4).

Many Canadians appear to still be struggling with meal and menu management for themselves and their households. Only 37.5% of Canadians believe that their ability to manage meals throughout the day has improved during the pandemic. Similarly, for snacks, only 31.5% of Canadians believe their ability to manage their snacks throughout the day has improved during the pandemic.

We also looked at sources of information, looking at two determinants, trust and influence. The most trusted source of information for Canadians when it comes to nutrition are online sources. Doctors come second at 39.3%, followed by friends and family members at 28.6% (Figure 5).

On the influence of food decisions, results are somewhat similar. The top choices were internet searches, followed by friends and family. Interestingly, health professionals such as nutritionists and dieticians ranked seventh, and Canada’s Food Guide only ranked eighth overall. No significant differences between generations were detected when looking at information sources (Figure 6). The survey also looked at gardening, which is another aspect of food literacy. Canadians are clearly embracing their time at home to vertically integrate and produce more food at home. A total of 51% of Canadians claim to have grown fruits or vegetables at home in 2020, and a total of 58% of Canadians intend to do the same in 2021, and 16% are not sure at this point.

Since food literacy is very much about well-being, the survey also looked at nutrition, physical fitness, and mental health. Results were telling of a struggling population. With nutrition, 36.2% of respondents believe they follow a healthier diet than before the pandemic. On physical fitness, 29.6% of respondents felt their physical fitness was better than before the pandemic. However, regarding mental health, scores were lower. Only 19.5% of respondents felt that their mental health was better than before the pandemic. That percentage was extremely low.

## 6. Discussion

Most studies of food literacy emphasise the acquisition of critical knowledge (information and understanding) over context. This evaluation looks at how COVID-19 impacted food literacy in an affluent country affected by the global pandemic. This survey provides interesting results since it captures how Canadians are impacted by the pandemic in the middle of a second wave, which has been more challenging than the first one. Unsurprisingly, through the lens of food literacy, health was the one driving factor that impacts most Canadians the most, 10 months into the pandemic.

The survey looked at health and well-being, and results show Canadians are not doing well, especially mentally [43]. This is likely since most have been emphasising their attention on the virus, and safety measures, to stay safe. Discussions on food autonomy have been prominent since the beginning of the pandemic, which may explain why most Canadians see it as a priority.

### 6.1. Increase in Food Literacy during Pandemic

Overall, food literacy-based metrics included in this exploratory survey did not significantly change because of the pandemic. Nonetheless, food literacy-based outcomes did somewhat improve. The average Canadian knows more recipes than before the pandemic, and some have used ingredients for the first time. Where Canadians seek information and what sources they trust have not changed immensely with the pandemic. The authors were expecting food literacy to have increased more due to more time spent at home and in the kitchen. Increasing confidence in a behaviour increases the potential for engagement and adaptation of the specific behaviour. Being forced to spend more time at home caused more Canadians to cook, but few changed habits, and it is unclear whether habits have changed permanently at this point. Food literacy improvements observed may provide positive coping skills to help reduce perceived risks and uncertainty in the Canadian population. The pandemic may have caused many Canadians to spend more time at home, yet the changes were not as important as anticipated.

The results of this study may suggest that higher levels of food literacy can be associated with more self-control, less impulsiveness, and healthier food consumption, as suggested by Lawlis et al. [1] and West [5]. As explained before, food literacy can give someone a sense of ownership, command of food access, and health. Notwithstanding mental health, a quarter of Canadians feel that their nutrition and physical health is better than before the pandemic. The results of this survey do not indicate, beyond a reasonable doubt, that Canadians are significantly more food literate as a result of COVID-19.

Food illiteracy occurs when households lack information on food or are not actively engaged with food systems, one way or another. However, it is not limited to only those households at the very bottom of the income distribution. Our research suggests that food illiteracy is not only a result of having insufficient information but also of lacking experience and information. These observations are very clear. Households that lack information on basic food concepts are more likely to experience food illiteracy. This is particularly the case for those who are younger and with lower levels of education and income. Given that the relationship between food literacy and security is strong [44], food literacy may be particularly important in helping low-income households cope with their limited resources. If this is the case, finding ways to help households better understand and manage their food stocks, learn recipes, and/or add new ingredients may help them to avoid food insecurity.

### 6.2. Practical and Policy Implications

Practical implications would include more information provided to consumers. In Canada, options abound when ordering food, and perhaps the temptation is there not to cook, learn recipes, or discover new ingredients [45]. Food companies, namely grocers, will need to think of ways to support consumers looking for ideas as they may be spending more time at home than before the pandemic. Policy implications are also numerous. Policies that are intended to address food literacy must attack the root causes of food illiteracy, and this exploratory research suggests that having more financial resources or education is not sufficient to avoid food illiteracy. During uncertain times, food illiteracy can arguably bring some comfort and a sense of control. Furthermore, it is not clear as to what the optimal number of recipes a consumer should know in order to be reasonably food literate. However, it is a metric that is fairly easy to measure by way of surveys. It should be monitored more closely over time to assess how Canadians are relating with food in their own kitchens. It will also be important to evaluate if any new ingredients are being adopted more than others by consumers cooking at home and the reasons why.

Additional research is needed to validate how global-scale events that generate a perception of uncertainty can impact food literacy over time. Further evaluations should look at different age groups, socioeconomic groups, and in different contexts.

This study has two limitations. The diversity of evidence collected here makes it difficult to combine the results of the studies to provide a conclusive conception of “food literacy” in uncertain times. Many households were affected by the pandemic financially, which also may have changed how they consumed food and perceived risks. This aspect was not thoroughly measured in this study. More research is necessary to better understand how pandemics, or any other global phenomenon, can influence food literacy. Furthermore, the low quality of evidence limits our ability to understand how food literacy may impact health and food security.

## 7. Conclusions

Measuring a nation’s food literacy is critical. The results are salient as they indicate a slight association between food literacy and risk perception in times of uncertainty. Many Canadians have made changes, but the numbers were not as high as anticipated in relation to cooking, although the number of new gardeners was impressive. The implications are that feelings of food insecurity and the fact that participants who experience a more domesticated lifestyle could impact food literacy. It may be necessary to further evaluate how food literacy can evolve over time beyond a pandemic or other major global event.

The COVID-19 pandemic has brought forward several unknowns. Yet what we do know is that the magnitude of the COVID-19 misfortune for the world demands that it be studied in detail in terms of key variables that impact food literacy. A more food literate market, as outlined in past literature, will have a very different relationship with food over time.

## Figures and Tables

**Figure 1 ijerph-18-05485-f001:**
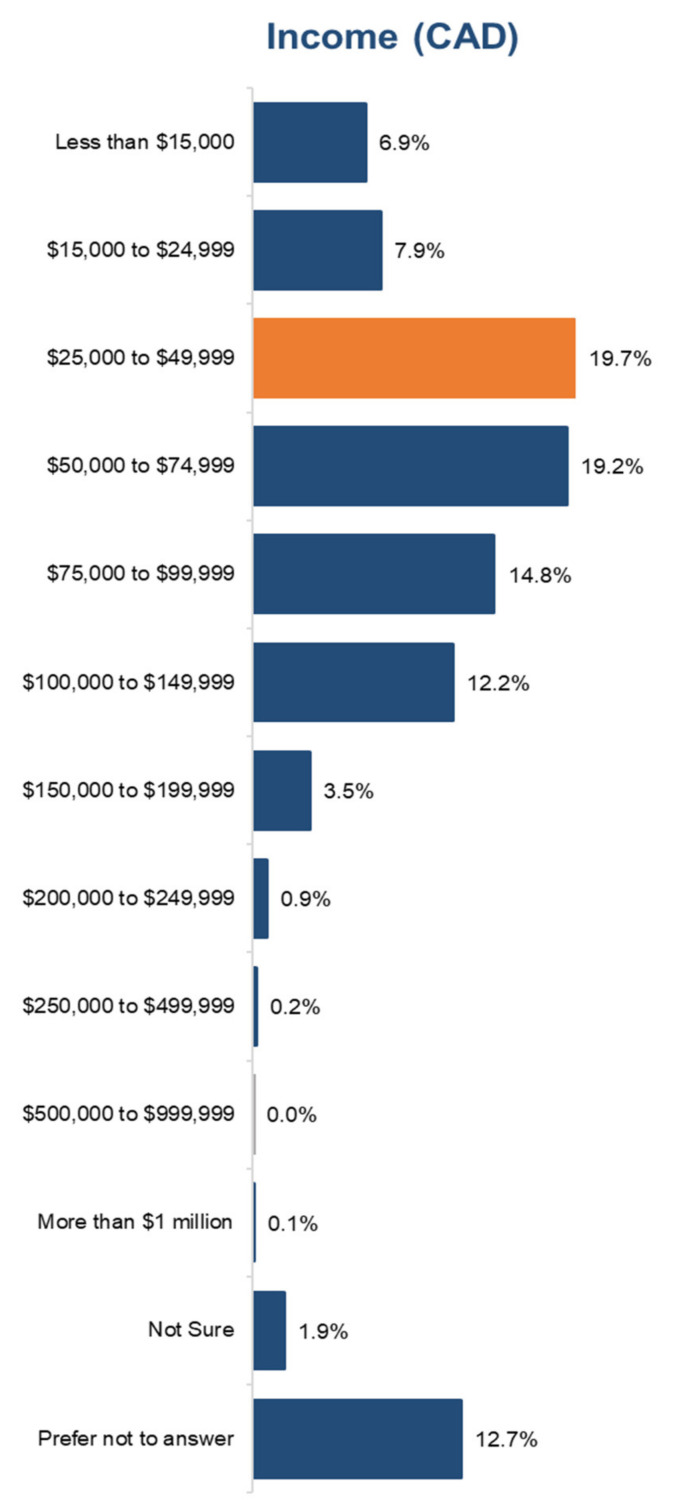
Respondents based on salaries.

**Figure 2 ijerph-18-05485-f002:**
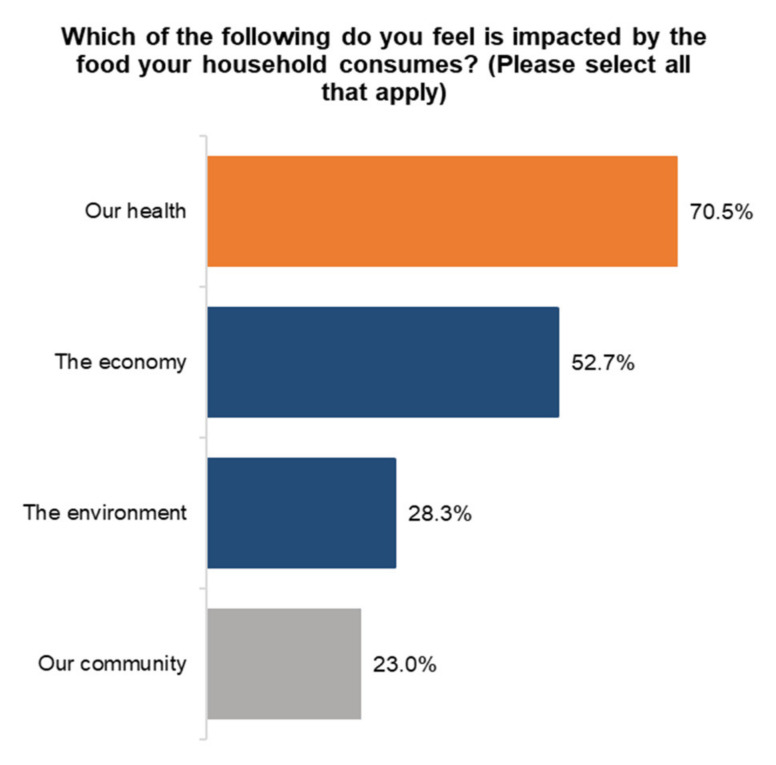
Influencing factors.

**Figure 3 ijerph-18-05485-f003:**
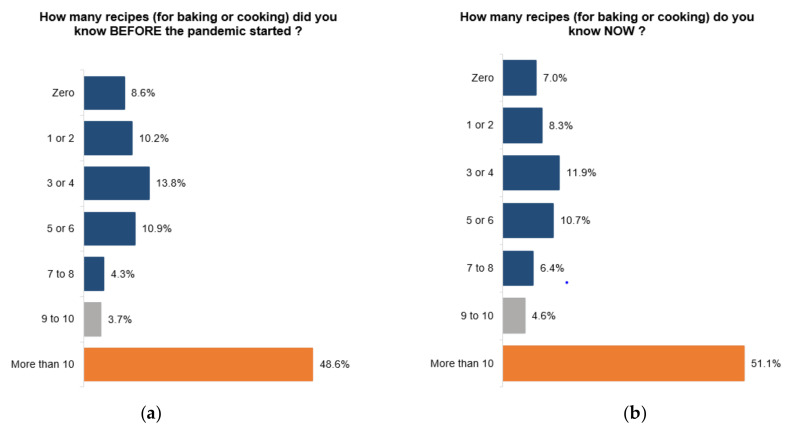
(**a**) Number of known recipes before pandemic, (**b**) Recipes known now.

**Figure 4 ijerph-18-05485-f004:**
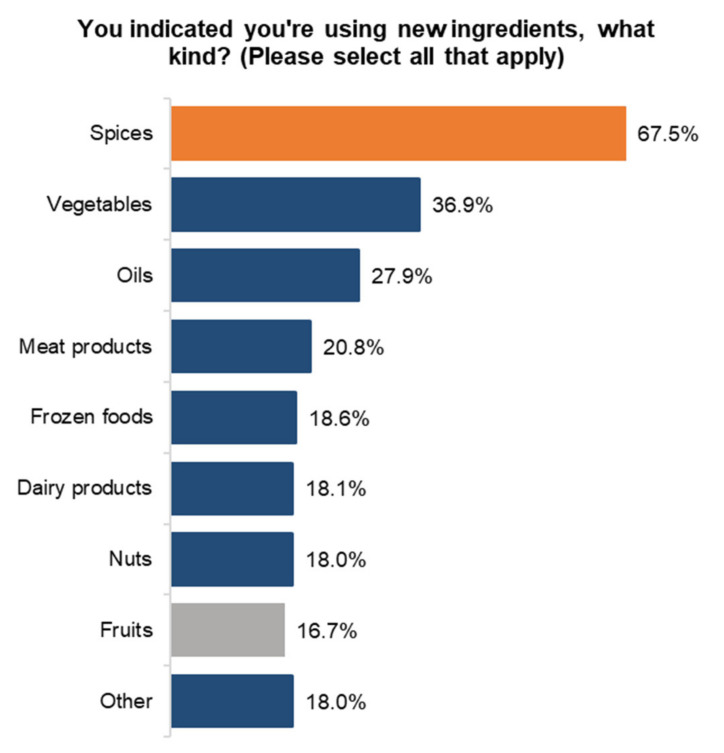
New ingredients used during pandemic.

**Figure 5 ijerph-18-05485-f005:**
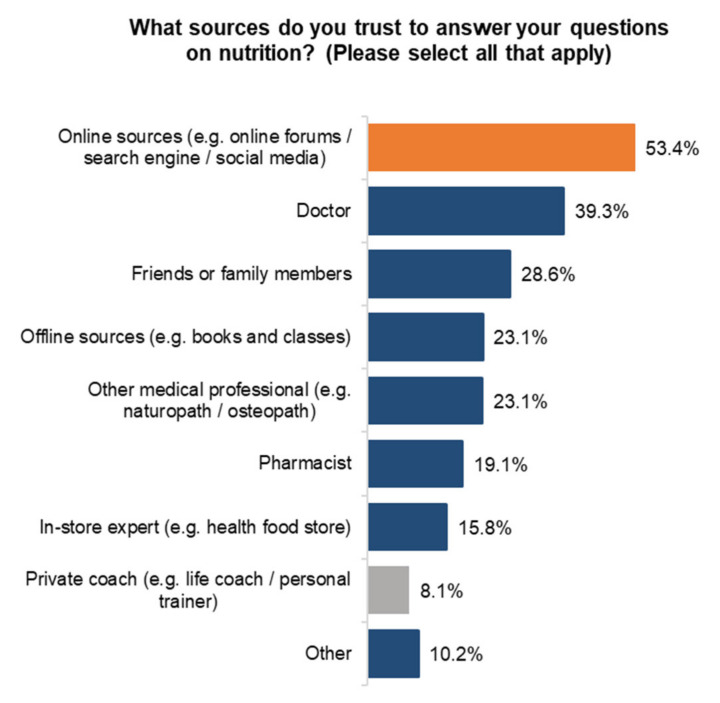
Nutrition sources of information used.

**Figure 6 ijerph-18-05485-f006:**
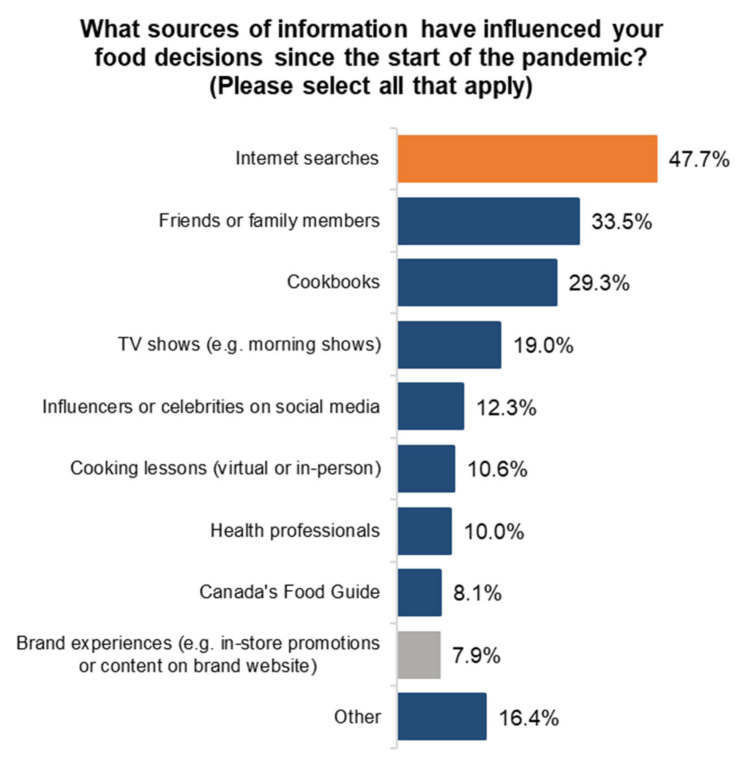
Sources of information for food choices.

**Table 1 ijerph-18-05485-t001:** Increase in number of recipes known.

Generations	Average Number of Recipes Known Before Covid-19	Average Number of Recipes Known Now
Gen z	4.7	5.6 (+0.9)
Millennials	4.9	6.0 (+1.1)
Gen x	6.2	6.7 (+0.5)
Boomers	7.4	7.6 (+0.2)
Canada	6.2	6.7 (+0.5)

## Data Availability

The data presented in this study are available on request from the corresponding author.

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
