# Peer review of "The Impact of COVID-19 on Canada’s Food Literacy: Results of a Cross-National Survey"

_ijerph, 2021, doi:10.3390/ijerph18105485_

Round 1

Reviewer 1 Report

This is a descriptive manuscript summarizing the concept of food literary and results of an exploratory survey that was conducted on food literacy during the COVID pandemic. Surveys like these will provide guidance to policy makers and educators to develop programs to support good eating habits in general as well as in times of crises. Thus, these are important results; however, several points should be addressed in order for the manuscript to be accepted. Specifically, this is a very descriptive report and including some statistical analyses would be beneficial. Other specific points are outlined below.  

Introduction: 

A definition of food literacy should be included early on (line 21) to better guide the reader (perhaps rearrange to move lines 30-31 up). It is unclear how food integrity (line 26) plays into food literacy (if at all) so in order to avoid confusion it would be advisable to include definitions. Similar with other concepts introduced in lines 28-30 (for example, food education which is first defined in lines 112-114). Perhaps even including a box with definitions could be beneficial. 

Lines 34-39: Might it also be worth mentioning that research on food literacy is scarce since the world has not seen a crisis like the COVID pandemic in a long time?  

There is some repetition throughout the introduction. For example, lines 37-38, lines 41-42, lines 58-60. This should be revised to clearly state the aim of the paper at the end of the introduction.  

Provide references for lines 52-53 and 60-61.  

Methods and Results: 

A table with information on demographics (age, gender, race, income, household size, etc) of the respondents should be included.   

As a general comment, would it be possible to include the questions that were used in the survey as a supplement? That way similar surveys could be designed in other countries and results might be able to be compared.  

Would it be possible to perform statistical analysis between some of the group differences described? For example, knowledge about recipes between different generations, differences by socioeconomic group? Perhaps the implications of these differences could be discussed in terms of policies/education (as started in lines 305 and below when discussing impact of education and income level on food literacy) 

It would also be interesting to see whether the different generations used different sources of information and the influences on food choices (lines 251-262). Perhaps a sub-analysis here is possible.  

Th authors mention that the amount of new recipes learned was lower than expected”. Please include what the expectations were and why.   

Some studies have shown that mental health was negatively affected during the pandemic. Where similar results (worsening of mental health perception during the pandemic (lines 269-271)) found in this survey? If mental health stayed the same in the majority of the population, then the conclusion that “Canadians are not doing well” (lines 280-281) would not follow. Especially since one wouldn’t expect mental health to get better if it is already good at baseline. Please include the distinction here. Similar for the other health outcomes. Perhaps describing the percentage of respondents who felt lie their physical activity, mental health, etc. has worsened during the pandemic would also be informative.  

Please include the rationale for why it was expected that food literacy would increase more that it has (lines 289-290) 

In lines 298-299 the authors discuss that “higher levels of food literacy can be associated with more self-control, less impulsiveness, and healthier food consumption. However, from the current manuscript it is not clear where these results are reported. Please revise.  

The age range for the different generations (Boomers, Generation X, etc.) should be included. 

Minor comments: 

Line 47: Should it be Winter 2020? 

Line 176: The amount of steps is missing. 

Line 345: should be brought  

Author Response

Report for Reviewer 1

Thank you for your insightful comments and suggestions. We have provided a response on a point-by-point basis.

This is a descriptive manuscript summarizing the concept of food literary and results of an exploratory survey that was conducted on food literacy during the COVID pandemic. Surveys like these will provide guidance to policy makers and educators to develop programs to support good eating habits in general as well as in times of crises. Thus, these are important results; however, several points should be addressed for the manuscript to be accepted. Specifically, this is a very descriptive report and including some statistical analyses would be beneficial. Other specific points are outlined below.  

Introduction

A definition of food literacy should be included early on (line 21) to better guide the reader (perhaps rearrange to move lines 30-31 up). It is unclear how food integrity (line 26) plays into food literacy (if at all) so to avoid confusion it would be advisable to include definitions. Similar with other concepts introduced in lines 28-30 (for example, food education which is first defined in lines 112-114). Perhaps even including a box with definitions could be beneficial. 

Authors: Good point. We have now added a definition of food literacy based on past literature. The “integrity” comment was a mistake. We meant “literacy”. That was changed as well.

Paragraph reads as follows: Food literacy is related to an individual’s capacity to make feasible food decisions which balance food needs using available resources (Charlebois et al., 2019; West, 2020). A person should be able to assess the quality of food options, including how impactful food choices are to the welfare and wellbeing of communities (Lawlis et al., 2019). 

When examining the discourse surrounding food literacy, cooking often materi-alizes as a dominant theme (Velardo, 2015). Embedded in the concept of food literacy is health and nutrition literacy which are topics that have been well researched over the years (West et al., 2020). In other words, food literacy is one aim of food education, others include kitchen-cooking literacy, nutrition literacy or health literacy (Raffoul and Hammond, 2018). Having food literacy means that a person is able to make prac-tical food decisions that balance food needs using accessible personal resources (Poelman et al., 2018).

Lines 34-39: Might it also be worth mentioning that research on food literacy is scarce since the world has not seen a crisis like the COVID pandemic in a long time?  

Authors: Indeed. We have added the following sentence to make the point clear. “This is probably because measures of food literacy used in a crisis context is seldom found together in datasets. The global coronavirus offers a unique food literacy context. To address this gap in the literature we collected data on a nationally representative sample of Canadians about both food literacy and how it effects financial literacy (Azevedo Perry et al., 2017).”

There is some repetition throughout the introduction. For example, lines 37-38, lines 41-42, lines 58-60. This should be revised to clearly state the aim of the paper at the end of the introduction.  

Authors: We have made minor changes to the text in that portion of the manuscript. The paragraph now reads as follows: When examining the discourse surrounding food literacy, cooking often materi-alizes as a dominant theme (Velardo, 2015). Embedded in the concept of food literacy is health and nutrition literacy which are topics that have been well researched over the years (West et al., 2020). In other words, food literacy is one aim of food education, others include kitchen-cooking literacy, nutrition literacy or health literacy (Raffoul and Hammond, 2018).

Provide references for lines 52-53 and 60-61.  

Authors: We have added Spring (2020) as a reference in that section.

Methods and Results

A table with information on demographics (age, gender, race, income, household size, etc) of the respondents should be included.   

Authors: We included one table for salaries, and we did add the information requested. The section reads as follows: The sample included 54% self-identified women versus 45% self-identified men. A total of 45% of respondents were Millennials (1981-1996), 34% were Gen Xs (1966-1980) and 15% were Boomers (1946-1965). A total of 59% of households had children in them. Information about salaries is shown in Figure X.  

As a general comment, would it be possible to include the questions that were used in the survey as a supplement? That way similar surveys could be designed in other countries and results might be able to be compared.  

Authors: We gladly added all of the main questions as a supplement, at the end of the article.

Would it be possible to perform statistical analysis between some of the group differences described? For example, knowledge about recipes between different generations, differences by socioeconomic group? Perhaps the implications of these differences could be discussed in terms of policies/education (as started in lines 305 and below when discussing impact of education and income level on food literacy)?  

Authors: Good point. We have added the following paragraph to make the policy implication portion more robust, as suggested: “Furthermore, it is not clear as to what the optimal number of recipes a consumer should know in order to be reasonably food literate, it is a metric that is fairly easy to measure by way of surveys. It should be monitored more closely over time to assess how Ca-nadians are relating with food in their own kitchens. It will also be important to eval-uate if any new ingredients are being adopted more then others by consumers cooking at home, and reasons why.”  

It would also be interesting to see whether the different generations used different sources of information and the influences on food choices (lines 251-262). Perhaps a sub-analysis here is possible.  

Authors: We did not see significant differences between generations which is why nothing was mentioned. We did add this line for clarity: “No significant differences between generations were detected when looking at information sources.”

Th authors mention that the amount of new recipes learned was “lower than expected”. Please include what the expectations were and why. 

Authors: It was more of an opinion. It was deleted.   

Some studies have shown that mental health was negatively affected during the pandemic. Where similar results (worsening of mental health perception during the pandemic (lines 269-271)) found in this survey? If mental health stayed the same in the majority of the population, then the conclusion that “Canadians are not doing well” (lines 280-281) would not follow. Especially since one wouldn’t expect mental health to get better if it is already good at baseline. Please include the distinction here. Similar for the other health outcomes. Perhaps describing the percentage of respondents who felt lie their physical activity, mental health, etc. has worsened during the pandemic would also be informative.  

Authors: Good comment. We have now added a baseline, a reference from the CMAJ (Murray, Terry. (2021). Unpacking "long COVID". Canadian Medical Association Journal (CMAJ), 193(9), E318-E319.)

Please include the rationale for why it was expected that food literacy would increase more that it has (lines 289-290).  

Authors: We added this comment to that section: “Authors were expecting food literacy to have increased more, due to more time spent at home and in the kitchen.”

In lines 298-299 the authors discuss that “higher levels of food literacy can be associated with more self-control, less impulsiveness, and healthier food consumption”. However, from the current manuscript it is not clear where these results are reported. Please revise.  

Authors: WE added this sentence to that section for clarity: Results in this study may suggest that higher levels of food literacy can be associ-ated with more self-control, less impulsiveness, and healthier food consumption as suggested by Lawlis et al. (2019) and West (2020).

The age range for the different generations (Boomers, Generation X, etc.) should be included. 

Authors: It is now included in the methodology section.

Minor comments: 

Line 47: Should it be Winter 2020? 

Line 176: The amount of steps is missing. 

Line 345: should be brought  

Authors: All of these have been changed. It is indeed Winter 2021 for the third wave.

Reviewer 2 Report

COVID-19 has focused on transmission, symptoms, structure and its structural proteins of SARS-CoV-2 as well as synthetic inhibitors, however functional food for treatment of human cells disease are crucial to combat COVID-19. This report reveals how COVID-19 impacted food literacy in a country affected by the global pandemic, especially Canadians have learned new recipes,  practical and policy implications are research directions. These results are of great theoretical significance and practical value for solving COVID-19. The analysis process is comprehensive, good organized, large amount of information and so on. Minor revision can be published in International Journal of Environmental Research and Public Health. However, there are some major issues need to be improved:

  1. Abstract: Further modifications to the value of prominent functional foods are required;
  2. Introduction: Updated reference to supplement functional food for treatment of  COVID-19,such as https://www.hindawi.com/journals/omcl/2020/3836172/
  3. Results: It's easier to understand different sections with subheadings;
  4. Discussion: The necessary references shall provide and compare and analyze the characteristics and innovation of the paper.

Author Response

Report for Reviewer 2

Thank you for your insightful comments and suggestions. We have provided a response on a point-by-point basis.

COVID-19 has focused on transmission, symptoms, structure and its structural proteins of SARS-CoV-2 as well as synthetic inhibitors, however functional food for treatment of human cells disease are crucial to combat COVID-19. This report reveals how COVID-19 impacted food literacy in a country affected by the global pandemic, especially Canadians have learned new recipes, practical and policy implications are research directions. These results are of great theoretical significance and practical value for solving COVID-19. The analysis process is comprehensive, good organized, large amount of information and so on. Minor revision can be published in International Journal of Environmental Research and Public Health. However, there are some major issues need to be improved:

Abstract: Further modifications to the value of prominent functional foods are required;

Authors: We did not quite understand the comment about functional foods. The study is not really about functional foods per se but about food literacy. Apologies. Some clarification would be required.

Introduction: Updated reference to supplement functional food for treatment of  COVID-19,such as https://www.hindawi.com/journals/omcl/2020/3836172/

Authors: Same for this comment. We were not sure what is being asked. Apologies.

Results: It's easier to understand different sections with subheadings.

Authors: We have added a few more subheading for better readability.

Discussion: The necessary references shall provide and compare and analyze the characteristics and innovation of the paper.

Authors: Thank you. We have made a few changes to the manuscript.

Round 2

Reviewer 1 Report

Thank you for addressing previous comments. A couple of comments remain:

It appears that the demographics table was not submitted.

Statisitcal analysis is not described in the manuscirpt. If this is a purely descriptive study, then it should be clearly stated and statistical terms such as "significant difference" or "correlation" should not be used. 

Author Response

Report (2) to Reviewer 1

Thank you again for your comments. Below are our responses.

Thank you for addressing previous comments. A couple of comments remain:

It appears that the demographics table was not submitted.

Authors: We added the following information. Please note that no data on race was collected for this survey.

“The sample included 54% self-identified women versus 45% self-identified men. A total of 45% of respondents were Millennials (1981-1996), 34% were Gen Xs (1966-1980) and 15% were Boomers (1946-1965). A total of 59% of households had children in them. Information about salaries is shown in Figure X. No data on race was collected for this survey.”  

Statistical analysis is not described in the manuscript. If this is a purely descriptive study, then it should be clearly stated and statistical terms such as "significant difference" or "correlation" should not be used.

Authors: We have deleted all references to statistical analysis (correlation). WE also made changes to the first sentence in the result section, making it clear results are descriptive.

“Given that food literacy can been a coping mechanism to offset these vulnerabilities, this exploratory survey garnered interesting descriptive results.”